# Preisach Elasto-Plastic Model for Mild Steel Hysteretic Behavior-Experimental and Theoretical Considerations

**DOI:** 10.3390/s21103546

**Published:** 2021-05-19

**Authors:** Dragoslav Sumarac, Petar Knezevic, Cemal Dolicanin, Maosen Cao

**Affiliations:** 1Department of Technical Sciences, Civil Engineering, State University of Novi Pazar, 36300 Novi Pazar, Serbia; pknezevic@np.ac.rs (P.K.); cdolicanin@np.ac.rs (C.D.); 2College of Civil and Architecture Engineering, Chuzhou University, Chuzhou 239000, China; 3Jiangxi Province Key Laboratory of Environmental Geotechnical Engineering and Hazards Control, Jiangxi University of Science and Technology, Ganzhou 341000, China; cmszhy@hhu.edu.cn; 4Nantong Ocean and Coastal Engineering Research Institute, Hohai University, Nantong 226000, China

**Keywords:** cyclic loading, elastic, plastic, locked-in energy, heat, cyclic counting, temperature and strain measurements

## Abstract

The Preisach model already successfully implemented for axial and bending cyclic loading is applied for modeling of the plateau problem for mild steel. It is shown that after the first cycle plateau disappears an extension of the existing Preisach model is needed. Heat dissipation and locked-in energy is calculated due to plastic deformation using the Preisach model. Theoretical results are verified by experiments performed on mild steel S275. The comparison of theoretical and experimental results is evident, showing the capability of the Presicah model in predicting behavior of structures under cyclic loading in the elastoplastic region. The purpose of this paper is to establish a theoretical background for embedded sensors like regenerated fiber Bragg gratings (RFBG) for measurement of strains and temperature in real structures. In addition, the present paper brings a theoretical base for application of nested split-ring resonator (NSRR) probes in measurements of plastic strain in real structures.

## 1. Introduction

The phenomenon of hysteresis has been a long-standing research interest. It occurs in various branches of physics: magnetism, adsorption, viscoelasticity, plasticity, etc. Its origin starts in 1935 after the paper published by Preisach [1] dealing with the problem of magnetism. After that the Presiach model extended to the problem of adsorption [2]. The first application was in mechanics, for solving problems of cyclic plasticity of axially loaded bars, which was elaborated in [3,4]. After that, the method was extended to elasto-plastic cyclic bending, as described in [5,6]. Low cycle fatigue due to earthquake loading was elaborated in [6]. The problem of the plateau associated with the mild steel is considered in [7].

In this paper theoretically and experimentally all phenomenon of elasto-plastic cyclic axial loading of the mild steel S275 are examined: Lüder dilatation, locked-in energy, dissipated heat, and temperature increase of the surface of specimens due to plastic deformation. For other types of mild, so called construction steels, such are S235 and S355, results can be found in [7].

The motivation of the authors for this paper is to use advantages of the Preisach model to make as simple as possible the calculation of stresses and strains in structures under extremely severe loadings. In addition, this paper has the intention to make connection with the scientists who are investigating reliable embedded sensors, which are beneficial for structural health monitoring (SHM).

To measure strains and temperature in real structures in civil and mechanical engineering and aeronautics, embedded sensors can be used, as explained in [8]. The purpose of this paper is to establish stress–strain relations and construction of cyclic stress–strain loops for arbitrary elements in structures. This is very important for structures resistant to earthquakes or bridges exposed to heavy moving loads. In addition, aircraft structures are usually in the situation that are exposed to cyclic loads during service life. Also, temperature rises due to plastic deformation can be calculated using the algorithm given in this paper. This temperature can be measured with embedded elements as shown in [8]. As stated in [8]: “For the tensile tests, a cyclic procedure was chosen, which allowed us to distinguish between the elastic and plastic deformation of the specimen. An analytical model, which described the elastic part of the tensile test, was introduced and showed good agreement with the measurements.” Readers of the paper are invited to see the explanation how to measure strains (elastic and plastic) given in reference [8] (Section 2.2 and Section 3). It is in detail explained theoretically in Section 2.2 as an application of FBG embedded in cast aluminum during a tensile test. The test was modeled as cyclic uniaxial tension with elastic and plastic strain. The expression connecting the change of the Bragg wavelength due to external strain is derived. Than this result is applied in Paragraph 3 where experimentally obtained results are discussed. During experiments elastic (reversible) and plastic (permanent) strain are measured using FBG with satisfactorily agreement with classical extensometer measurements.

In addition, the present paper establishes a theoretical background, confirmed by experimental investigation, for application of a nested split-ring resonator (NSRR) probe in measurements of plastic strain in real structures [9]. Explanation of the application of this sensor, which was used for the first time to measure elastic and plastic strains in the reinforcing bar (rebar) of a simple supported reinforced concrete beam is explained in [9]: “The system utilizes electromagnetic coupling between the transceiver antenna located outside the beam, and the sensing probes placed on the reinforcing bar (rebar) surface inside the beam. The probes were designed in the form of a nested split-ring resonator, a metamaterial-based structure chosen for its compact size and high sensitivity/resolution, which is at μm/microstrains level. Experiments were performed in both the elastic and plastic deformation cases of steel rebars, and the sensing system was demonstrated to acquire telemetric data in both cases.” In Figure 9c of the paper [9] it can be seen that a plateau in the stress–strain diagram appears. This phenomenon is thoroughly examined in our paper.

## 2. Theoretical Model

The Preisach model, originally developed in 1935 to describe hysteresis in ferromagnetism, was named after its author [1]. This model represents the mapping of the input data functions into the output data function. It is based on the elementary nonlinear hysteresis operator *G_α,β_*, which is a discontinuous operator with local memory. However, by superposing these operators within the domain *Γ*, the Preisach hysteresis operator $\hat{\Gamma }u\left(t\right)$ is formed as a continuous system of infinitely many elementary operators connected in parallel (or in series):(1)$$f\left(t\right)=\hat{\Gamma }u\left(t\right)=\underset{\alpha \geq \beta }{\iint }G_{\alpha ,\beta }u\left(t\right)\mu \left(\alpha ,\beta \right)d\alpha d\beta$$
where $\mu \left(\alpha ,\beta \right)$ represents the Preisach weight function, according to which the elementary operators are arranged, and *G_α,β_* is the elementary hysteresis operator shown in Figure 1.

Structural mild steel behavior under monotonic loading is characterized by a phenomenon called the Lüders band, whose main feature is the development of a horizontal yield plateau. Those phenomena occur during the transition from the elastic region to the region of nonlinear plastic hardening.

The values of the upper and lower yield limits, and the Lüder dilatation, depend on a large number of parameters, such as the grain size of the steel [10], the rate of deformation, and the carbon content [11].

This local instability is characteristic only of monotonic loading, while the horizontal yield plateau disappears due to reversible loading.

### 2.1. Single Crystal Preisach Model

In this paper, a new type of Preisach model for response of structural mild steel under constant cyclic loading will be developed. Its basis is a model that describes the behavior of this type of steel under monotonic loading [7].

The behavior of the steel S275, under monotonic loading, is characterized by an unstable elastoplastic transition which reduces their workability and ductility. This phenomenon is a result of the separation of the free atoms (usually carbon or nickel) and their incorporation at the sites of existing and newly formed dislocations within the lattice formed by atoms of iron [12].

The initial part of the stress–strain diagram of these steel is linear and proportional to the modulus of elasticity *E* until the stress *σ_uy_* is reached This limit is called the upper yield stress. Reaching a given value is succeeded by a sudden drop in stress to the value of the lower yield point *σ_ly_*. This is followed by the formation of a yield plateau, with an approximate magnitude of 1–3% of the total dilatation, according to [13]. The cause of this behavior of the mentioned steel lies in the microstructure and atomic lattice. Free carbon and nitrogen atoms surround the dislocation, causing a high level of stress required to move it within the crystal lattice *σ_uy_*. After initiating the dislocation, its motion continues relatively easily, forming a yield plateau, until a new regrouping of atoms within the crystal lattice occurs.

This phenomenon is called the Lüders band, and the length of the yield plateau is the Lüder dilatation *ε_L_*. The Lüders band represents a local, inhomogeneous deformation, which causes the development of a yield plateau until the Lüders band spreads over the entire sample, followed by material hardening and homogeneous deformation (Figure 2).

The upper yield stress value *σ_uy_*, of the mild construction steel, is very sensitive to small stress concentrations, alignment of the sample inside the machine jaws, as well as to other parameters. For this reason, the upper yield strength is neglected, and the value of the stress at which the transition from the elastic to the plastic region occurs is taken to be the value of the lower yield strength *σ_ly_*.

The basic principle in modeling elastoplastic material behavior is based on defining an analog mechanical model, determined by an appropriate set of algebraic and/or differential equations. A material model, describing the response of the subject steel under monotonic axial stress, is constructed by modifying the existing three-element model [3,4,5,6]. By introducing a delay element, the delay of material hardening is achieved after reaching the yield strength *Y* = *σ_ly_*.

The concavity of the *σ*-*ε* curve in the hardening zone is achieved by its approximation with three lines, which makes the working diagram quintuple linear. This is achieved by introducing additional elements into the three linear mechanical model, as shown in the Figure 3.

The material properties of the mechanical model, shown in Figure 3b, are defined by Equation (2).
(2)$$E=E_{0}\left(L_{0}+L_{1}+L_{2}\right)/L_{0}\;E_{a}=E_{0}\times E_{1}/\left(E_{0}+E_{1}\right)\;E_{1}=h_{1}\left(L_{0}+L_{1}+L_{2}\right)/L_{1}\;E_{h}=E_{0}\times E_{1}\times E_{2}/\left(E_{0}+E_{1}+E_{2}\right)\;E_{2}=h_{2}\left(L_{0}+L_{1}+L_{2}\right)/L_{2}$$

It is possible to define a new hysteresis mechanical model based on the working diagram shown in Figure 3, which describes the behavior of a structural mild steel single crystal under monotonic loading:(3)$$\sigma (t)=\frac{E}{2}\iint_{\alpha \geq \beta }\delta \left(\alpha -\beta \right)G_{\alpha ,\beta }\varepsilon (t)d\alpha d\beta -\frac{E}{2}\iint_{\alpha \geq \beta }\delta \left(\alpha -\beta -2\varepsilon_{1}\right)G_{\alpha ,\beta }\varepsilon (t)d\alpha d\beta +\;\frac{E_{a}}{2}\iint_{\alpha \geq \beta }\delta \left(\alpha -\beta -2\varepsilon_{2}\right)G_{\alpha ,\beta }\varepsilon (t)d\alpha d\beta +\frac{E_{h}-E_{a}}{2}\iint_{\alpha \geq \beta }\delta \left(\alpha -\beta -2\varepsilon_{3}\right)G_{\alpha ,\beta }\varepsilon (t)d\alpha d\beta -\;\frac{E_{h}}{2}\iint_{\alpha \geq \beta }\delta \left(\alpha -\beta -2\varepsilon_{4}\right)G_{\alpha ,\beta }\varepsilon (t)d\alpha d\beta .$$

And its appropriate Preisach function:
(4)$$\mu (\alpha ,\beta )=\frac{1}{2}\frac{\partial^{2}f_{\alpha ,\beta }}{\partial \alpha \partial \beta }=\frac{E}{2}\times \left[\delta (\alpha -\beta )-\delta (\alpha -\beta -2\varepsilon_{1})\right]+\frac{E_{a}}{2}\times \delta (\alpha -\beta -2\varepsilon \;+\frac{E_{h}-E_{a}}{2}\delta \left(\alpha -\beta -2\varepsilon_{3}\right)-\frac{E_{h}}{2}\times \delta \left(\alpha -\beta -2\varepsilon_{4}\right).$$

The formation of a horizontal yield plateau and a phenomenon called the Lüders band represent the characteristic behavior of mild steels under monotonic loading [14]. Under the first load direction switch, a vanishing of the yield plateau occurs, with a complete transformation of the material behavior into an ideal elastoplastic behavior with hardening.

The cyclic behavior of the examined steel types is characterized by the formation of regular hysteresis loops, with no flattening. A new hysteresis model is established by excluding the yield plateau and modifying the material parameters.

A satisfactory approximation of hysteresis loops is possible using a three-element model [3]. In order to better match the real response of the material, a five-element model with a trilinear working diagram, shown in Figure 4, was used in this paper.

The material properties of the mechanical model, shown in Figure 4b, are defined by Equation (2), where substitution should be made:(5)$$E_{a}=E_{a}^{c}\;and\;E_{h}=E_{h}^{c}$$

It is possible to define a new hysteresis mechanical model based on the working diagram shown in Figure 4, which describes the behavior of a structural mild steel single crystal under cyclic loading:(6)$$\sigma^{c}(t)=\frac{E}{2}\iint_{\alpha \geq \beta }\delta \left(\alpha -\beta \right)G_{\alpha ,\beta }\varepsilon (t)d\alpha d\beta +\frac{E_{a}^{c}-E}{2}\iint_{\alpha \geq \beta }\delta \left(\alpha -\beta -2\varepsilon_{1}^{c}\right)G_{\alpha ,\beta }\varepsilon (t)d\alpha d\beta +\;\frac{E_{h}^{c}-E_{a}^{c}}{2}\iint_{\alpha \geq \beta }\delta \left(\alpha -\beta -2\varepsilon_{2}^{c}\right)G_{\alpha ,\beta }\varepsilon (t)d\alpha d\beta .$$

By switching to the single integral, where the integration of the first part on the right hand side of Equation (6) takes part along line β=α, second along line $\beta =\alpha -2\varepsilon_{1}^{c}$, and third along line $\beta =\alpha -2\varepsilon_{2}^{c}$ Equation (6) for stress becomes:
(7)$$\sigma^{c}(t)=\frac{E}{2}\int_{-\varepsilon_{s}}^{\varepsilon_{s}}G_{\alpha ,\alpha }\varepsilon (t)d\alpha +\frac{E_{a}^{c}-E}{2}\int_{2\varepsilon_{1}^{c}-\varepsilon_{s}}^{\varepsilon_{s}}G_{\alpha ,\alpha -2\varepsilon_{1}^{c}}\varepsilon (t)d\alpha +\frac{E_{h}^{c}-E_{a}^{c}}{2}\int_{2\varepsilon_{2}^{c}-\varepsilon_{s}}^{\varepsilon_{s}}G_{\alpha ,\alpha -2\varepsilon_{2}^{c}}\varepsilon (t)d\alpha$$

The first integral of Equation (7) represents the stress due to elastic deformation. The second and third addends describe the behavior of the material after reaching the yield strengths $Y_{1}^{c}$ and $Y_{2}^{c}$, respectively.

### 2.2. Polycrystal Preisach Model

To model the real polycrystalline material, such as mild steel S275, parallel or series connections of infinitely many single crystal elements (Figure 5) are exploited as in [15]. In this paper we are using parallel connections of monocrystalline elements with the range of the yield stress *Y_i_^min^ ≤ Y_i_ ≤ Y_i_^max^*. Then the stress in the polycrystalline element reads:(8)$$\sigma (t)=\int_{Y_{i}^{c,\min}}^{Y_{i}^{c,\max}}p(Y_{i}^{c})\sigma (Y_{i}^{c},t)dY_{i}^{c}$$

In Equation (8) *p(Y_i_)* is the yield strength probability density function. Assuming that the yield limits *Y_1_* are the same in all parallel-connected individual units and defining that the distribution functions of other *Y_i_* values are uniform, as in papers [3,4,5,6,7].
(9)$$p(Y_{i}^{c})=\frac{1}{Y_{i}^{c,\max}-Y_{i}^{c,\min}}=const$$

The total stress, due to strain as an input, becomes:(10)$$\sigma^{c}(t)=\frac{E}{2}\int_{-\varepsilon_{s}}^{\varepsilon_{s}}G_{\alpha ,\alpha }\varepsilon (t)d\alpha +\frac{E_{a}^{c}-E}{E}p\left(Y_{1}^{c}\right)\int_{Y_{1}^{c,\min}}^{Y_{1}^{c,\max}}\int_{2\varepsilon_{1}^{c}-\varepsilon_{s}}^{\varepsilon_{s}}G_{\alpha ,\alpha -2\varepsilon_{1}^{c}}\varepsilon (t)d\alpha dY_{1}^{c}+\frac{E_{h}^{c}-E_{a}^{c}}{E}p\left(Y_{2}^{c}\right)\int_{Y_{2}^{c,\min}}^{Y_{2}^{c,\max}}\int_{2\varepsilon_{2}^{c}-\varepsilon_{s}}^{\varepsilon_{s}}G_{\alpha ,\alpha -2\varepsilon_{2}^{c}}\varepsilon (t)d\alpha dY_{2}^{c}$$

Since the first addend of Equation (10) does not depend on $Y_{i}^{c}$, on the basis of the second it is $\alpha -\beta =2\varepsilon_{1}^{c}$, and on the basis of the third $\alpha -\beta =2\varepsilon_{2}^{c}$. The term *β* can be reintroduced into the equation, with shifts $-d\beta \times \left(E/2\right)=dY_{1}^{c}$, and $-d\beta \times \left(E_{a}^{c}/2\right)=dY_{2}^{c}$, where the negative sign of the shift is lost due to the change of integration boundaries within triangles:(11)$$\sigma^{c}(t)=\frac{E}{2}\int_{-\varepsilon_{s}}^{\varepsilon_{s}}G_{\alpha ,\alpha }\varepsilon (t)d\alpha +\frac{E\left(E_{a}^{c}-E\right)}{4}p\left(Y_{1}^{c}\right)\iint_{A^{\prime }}G_{\alpha ,\beta }\varepsilon (t)d\alpha d\beta +\frac{E_{a}^{c}\left(E_{h}^{c}-E_{a}^{c}\right)}{4}p\left(Y_{2}^{c}\right)\iint_{B\prime }G_{\alpha ,\beta }\varepsilon (t)d\alpha d\beta .$$

The integration domains in Equation (11) represent the areas of the bands between the corresponding lines in a bounded triangle (Figure 6), because the Preisach function exists only in these domains and otherwise is zero. Domain A′ represents the area between the lines $\alpha -\beta =2\varepsilon_{1}^{c,full}$, and $\alpha -\beta =2\varepsilon_{1}^{c,init}$, while domain B′ represents the area between the lines $\alpha -\beta =2\varepsilon_{2}^{c,full}$ and $\alpha -\beta =2\varepsilon_{2}^{c,init}$.

The Preisach function for polycrystalline internal hysteresis loops is defined as:(12)$$\mu^{c}(\alpha ,\beta )=\frac{E}{2}\delta \left(\alpha -\beta \right)+\frac{E\left(E_{a}^{c}-E\right)}{4}\left[H\left(\alpha -\beta -2\varepsilon_{1}^{c,init}\right)-H\left(\alpha -\beta -2\varepsilon_{1}^{c,full}\right)\right]\;+\frac{E_{a}^{c}\left(E_{h}^{c}-E_{a}^{c}\right)}{4}\left[H\left(\alpha -\beta -2\varepsilon_{2}^{c,init}\right)-H\left(\alpha -\beta -2\varepsilon_{2}^{c,full}\right)\right].$$

The geometric interpretation of Equation (12) represents the Preisach triangle shown in Figure 6.

### 2.3. Wiping Out and Congruency Properties of Preisach Model

Using the geometric interpretation of the Preisach model, the calculation of the double integral in Equation (11) can be avoided. Determination of the integral value comes down to the estimation of the area inside the Preisach triangle. Observing the elementary operators values, it can be stated that at any time, the Preisach triangle consists of points at which elementary operators are at a “switch on” position and points at which they are at a “switch off” position. It can be noticed that the value of the output, at some arbitrary moment, depends on the division of the boundary triangle into positive and negative sets *A+* and *A−*. Increasing and decreasing of the input leads to a change in this redistribution and the formation of a staircase line *L(t)*, which represents the boundary between these sets.

The vertices of the staircase line are the extreme values of the input data so the *L(t)* line represents the memory of this operator. The memory formation in Preisach model is achieved by changing the shape of the staircase line *L(t)*.

The wiping-out property of Preisach model can be described through its geometric interpretation. This phenomenon characteristic is that each local maximum erases the vertices of the line *L(t)* whose *α* coordinates are below that maximum, and each local minimum erases vertices whose *β* coordinates are above that minimum (Figure 7). Only the alternative values of the dominant extremes of the input data are stored in the Preisach model, while all others are deleted.

The Preisach model, therefore, possesses selective memory. The elementary hysteresis operator (Figure 1) has local memory, but by combining a large number of these operators, non-local model memory is achieved. This memory does not depend on the entire load history, but only on the dominant values of the input extremes, due to the wiping-out property.

Congruence is another fundamental property of the Preisach model, which can be easily demonstrated through the geometric interpretation of the model (Figure 8). If we observe two states, defined by Preisach triangles *T*_1_ and *T*_2_, in which the input varies in the same range, it can be noticed that the changes in the areas of the triangles are equal in both cases (Δ*T*_1_ = Δ*T*_2_), regardless of previous load histories. It follows that the output increments for two states with different previous load histories are mutually equal Δ*f*_1_ = Δ*f*_2_. The consequence of this property is that all inner loops, created by varying the input between the same dominant extremes, have the same shape and size, but a different position within the main loop, which depends on the previous load history. The complete matching of these loops could be achieved by their translation in the *f*-axis direction.

The wiping-out and congruence properties represent necessary and sufficient conditions for hysteresis nonlinearity to be defined by the Preisach model [16]. These phenomena are essential for further consideration of energy losses achieved during random load histories.

### 2.4. Heat Loses

The term hysteresis is mainly related to the appearance of the hysteresis loop. However, hysteresis is also associated with loss of energy manifested through energy losses. Energy losses accomplished during the formation of hysteresis loops were first considered in electromagnetism. The hysteresis energy losses in magnetism are defined by Charles Proteus Steinmetz as the surface of the hysteresis loop [17].

Thanks to this observation, the determination of hysteresis losses at cyclic loads is based on the principle of energy conservation. The analysis of the energy consumed in the formation of hysteresis is important for the field of continuum mechanics.

Determination of energy losses during the formation of hysteresis loops, for an arbitrary history of excitation, can be performed on the basis of Preisach hysteresis operator and its weight function *μ(α, β)*. The general solution was defined by Mayergoyz [16], while the analysis of energy dissipation in the plastic domain for a three-element model of materials is presented in the paper [3].

The overall plastic work *W_p_*, under cyclic loading, is spent on thermal changes *S*, and the energy remains trapped inside the material [4], called locked-in energy *W_L_*.
(13)$$W_{p}=S+W_{Ls}$$

If the heat losses *S* are defined using the Preisach hysteresis model, the total loss can be obtained as the sum of the losses in elementary hysteresis operators. The realized loss in one elementary hysteresis operator *G_α,β_*, at full cycle, as shown in Figure 1, is:(14)$$s_{cycle}=2\left(\alpha -\beta \right)$$
while the energy loss during one input change (switch up or switch down) is:(15)s=α−β

Since the product *μ*(*α*, *β*) × *Gα*, *β* can be considered as a rectangular loop with an output value ± *μ* (*α*, *β*), the energy loss in such a loop, during one input change, is defined as the product *μ*(*α*, *β*) (*α* − *β*). Whereas the Preisach hysteresis model is a set of a large (infinite) number of elementary hysteresis operators, the total energy losses are obtained by summing the energy losses of all elementary hysteresis operators. This is achieved by integration over the domain Ω where the value of the input within the Preisach triangle has changed. Energy losses at the cross-sectional level are then defined as:(16)$$S=\iint_{\Omega }\left|\mu \left(\alpha ,\beta \right)\right|\left(\alpha -\beta \right)d\alpha d\beta$$

The double integral in Equation (16) is determined as the volume with basis area Ω in the *α* − *β* plane and height in the z-direction with the value of (*α* − *β*). From Equation (16), to find the fraction of the plastic work dissipated into heat, one should integrate the Preisach function for a chosen model within the area of the limiting triangle. Following the procedure explained in [3] for a parallel connection of infinitely many slip and sliding elements connected in series one obtains:
(17)$$S=\frac{1}{3}\left(2Y_{\min}+\sigma +E\times \varepsilon_{p}\right)\times \varepsilon_{p}$$

For mild steel material constants that are used in the above equation are: *E* = 210 GPa, *Y*_min_ = 16 KN/cm^2^, σ = *Y*_max_ = 2 *Y*_min_ = 32 KN/cm^2^ and *ε_p_* = 1.25%.

Then from Equation (14) it is obtained for one monotonic cycle:(18)$$S=13,600\;\mathrm{kNm}/m^{3}=1.36\;\mathrm{kNcm}/{\mathrm{cm}}^{3}$$

One can easily calculate, following the procedure given in [3], the amount of plastic work *W_p_* as:(19)$$W_{p}=\int_{0}^{\varepsilon_{p}}\sigma d\varepsilon_{p}=\frac{1}{3}\left(Y_{min}+2\sigma +\frac{1}{2}E\varepsilon_{p}\right)\varepsilon_{p}$$

Finally, the difference is:(20)$$W_{L}=W_{p}-S=\frac{1}{3}\left(\sigma -Y_{min}-\frac{1}{2}E\varepsilon_{p}\right)\varepsilon_{p}$$

This represents the locked-in energy stored during the primary loading from initial, zero state stress to the state of stress *Y_min_ ≤ σ ≤ Y_max_*.

## 3. Temperature Field in Cyclically Loaded Cylindrical Specimen in Plastic Region

The Preisach model is capable to determine heat generation in the material during cyclic loading in the plastic region. Based on Equation (16), heat generated in specimen *S* is given by Equation (17). *S* is the volumetric source of heat and it is measured in Wh/m^3^, with the conversation factor: 1 Wh = 3.6 kNm.

From the description of the experiment given in Section 5, it can be seen that the loading, even cyclic, is slow and the strain rate is not considered. The temperature field is in the steady state condition. Let us consider the temperature field within the cylinder. For this purpose we begin with a sketch of a cylindrical section with a radius *r* and of width Δ*r*, and extended the entire length *L* of the cylinder as shown in Figure 9 and Figure 10. The energy balance for the cylindrical shell under steady state condition is [18,19]:(21)$$Q\left(r\right)+2\pi \times r\times \Delta r\times L\times S=Q\left(r+\Delta r\right)$$

Rearranging the above equation, dividing by Δ*r* and taking the limit as Δ*r* tends to zero, leads to:(22)$$\underset{\Delta r\to 0}{lim}\frac{Q\left(r+\Delta r\right)-Q\left(r\right)}{\Delta r}=2\pi rLS$$

Or:(23)$$\frac{dQ}{dr}=2\pi rLS$$

The heat flow rate can be expressed as:(24)$$Q=q_{r}A=q_{r}2\pi rL$$

Taking Fourier’s law, with the *k* as conductivity constant,
(25)$$q_{r}=-k\frac{dT}{dr}$$

leads to:
(26)$$Q=-2\pi rLk\frac{dT}{dr}$$

Substituting (23) into (20) one obtains:(27)$$\frac{1}{r}\frac{d}{dr}\left(r\frac{dT}{dr}\right)=-\frac{S}{k}$$

Multiplying by *r*, integrating once with respect to *r* and finally dividing by *r* leads to:(28)$$\frac{dT}{dr}=-\frac{S}{2k}r+\frac{C_{1}}{r}+C_{2}$$

After one more integration the temperature field is:(29)$$T\left(r\right)=-\frac{S}{4k}r^{2}+c_{1}lnr+C_{2}$$

In order that the temperature should be of finite value at *r* = 0 leads to C_1_ = 0. Then with the integration constant *C*_2_ we can find from the boundary condition that for *r = R, T(R) = T_s_:*(30)$$C_{2}=T_{s}+\frac{S}{4k}R^{2}$$

Substituting (30) into (29) we obtain the final temperature distribution:(31)$$T\left(r\right)=T_{s}+\frac{S}{4k}\left(R^{2}-r^{2}\right)$$

The maximum temperature is at the center of the cylinder (specimen), as is expected for *r = 0*:(32)$$T\left(r\right)=T_{s}+\frac{S}{4k}R^{2}$$

The most important for us is to theoretically determine the surface temperature *T_s_*. Total heat generated within the cylinder, which is the product of heat generated rate per unit volume *S* and the volume of the cylinder, will be transferred to the surrounding area. On the other hand, the total heat transferred to the outside of the cylinder is the product of the heat flux as convective heat transfer and the surface area of the cylinder:(33)$$Q=\pi R^{2}LS=h_{0}\left(T_{s}-T_{0}\right)2\pi RL$$

From the above equation, finally we obtain:(34)$$T_{s}=T_{0}+\frac{R}{2h_{0}}S$$

Usual values for the conduction coefficient *k* for steel are *k* = 16–24 W/mK, and the convection coefficient *h*_0_ = 10–100 W/m^2^K.

Substituting the value of heat generation given by (18) into (34), taking for *h*_0_ = 25 W/m^2^K, one obtains.
T = 22 + 3.777 = 25.777 °C(35)

## 4. Preliminary Analysis of Load History

The Preisach model presents a very powerful tool in the analysis of the cyclic behavior of ductile materials and accompanying energy phenomena. However, loading in real structures in civil, mechanical and aerospace engineering is mainly characterized by a random load history. Solving complicated problems is simplified by prior analysis of the load history using some of the methods for counting cycles within a random load history.

Several methods for determining the number of cycles have been developed, where each has found its application in solving a certain type of problem. The purpose of each method is to display random load histories through a set of load histories with constant amplitudes. It should be noted that before applying any cycle counting method, load histories are filtered so that only values of local extremes are retained from real load histories. In general, all existing methods can be classified into three groups [20,21,22]:
–Peak counting methods–Range counting methods–Level crossing methods.

The cyclic load’s amplitude is usually a random variable. The rainflow method is the most common method of determining the number of cycles of such a load. Its prevalence and acceptance are reflected in the physical nature of the full cycle, which is represented by a closed hysteresis loop on the stress–strain diagram.

The concept of this method was first given by Matsuishi and Endo [23], where an analogy was made with raindrops, which flow through a row of canopy roofs. This method is developed for counting half-cycles and then pairing them into full cycles (range counting method).

This algorithm requires knowledge of the entire load history, which does not allow immediate data processing and requires large memory for storing the same. To eliminate these shortcomings, several algorithms have been developed, where the most well-known are:
-three-points algorithm (Figure 11a)
(36)$$Y=\left|P_{i+1}-P_{i}\right|\leq X=\left|P_{i+2}-P_{i+1}\right|$$
wherein *i* = 1, 2, 3, … *M* − 2, and *M* is the length of the considered data series, and-four-points algorithm (Figure 11b)
(37)$$Z=\left|P_{i+1}-P_{i}\right|\geq Y=\left|P_{i+2}-P_{i+1}\right|\leq X=\left|P_{i+3}-P_{i+2}\right|$$
wherein *i* = 1, 2, 3, … *M* − 3, and *M* is the length of the considered data series.

These algorithms and their variations are based on the current analysis of the load history and the extraction of full cycles from it, where the full cycle manifests as a closed hysteresis loop.

Modification of the four-points criterion is possible by adding *n* points to the calculation algorithm, achieving a higher processing speed of the load history, but also requiring a larger memory for data storage. The newly defined algorithm (Figure 12) is a 4 + *n* points criterion, which is implemented if the 4 points criterion is not met.

No matter which the rainflow method algorithm is applied, the residuum of the uncounted half cycles inevitably appears. Methods of processing these residues are presented in [24].

The rainflow method is a nonlinear numerical algorithm, not a mathematical function [25]. The use of rainflow algorithms is limited to post-processing the results. Since the algorithms of this method require a series of data (points) and not only instantaneous measurements, this method is not suitable for real-time data processing and is more often used as a post-processing tool.

The Preisach hysteresis operator, as a continuous rate-independent operator, has found its application in the assessment of material damage depending on the number of load cycles. Its purely mathematical form allows its application in real-time, unlike the rainflow method which is limited to subsequent data processing. The results comparison of these two approaches is shown in [26,27], confirming the compatibility between the rainflow method and the Preisach hysteresis operator. The results compatibility of these two different approaches in determining the number of cycles in a random load history is shown by Equation (36) towards estimating fatigue damage.
(38)$$D_{ac}\left(s\right)=\frac{c\left(s\right)\left(\alpha ,\beta \right)}{N\left(\alpha ,\beta \right)}=Var\left(P\left(\alpha ,\beta \right)\right)$$
where *c*(*s*)(*α*,*β*) represents the cycle number determined by the rainflow counting method, *N*(*α*,*β*) is the number of cycles up to failure and *Var*(*P*(*α*,*β*)) is the variation of the Preisach function up to failure under cycles *N*(*α,β*).

The calculation of energy losses *S* and trapped energy would be greatly simplified by extracting hysteresis loops of the same dimensions from random load histories. The rainflow counting method in principle identifies the ranges, which correspond to closed hysteresis loops.

## 5. Experimental Results and Model Verification

The experiments presented in the paper were conducted on samples of structural mild steel class S275 at room temperature.

All experimental samples are cylindrical, with a total length of 190 mm and a circular cross-section of the measuring part, with a diameter of d = 10 mm. The dimensions of the test specimens are shown in Figure 13.

The samples were prepared according to the standard [28]. The influence of surface roughness was eliminated by the surface finishing of the sample.

Experimental equipment consisted of the SHIMADZU Servo Pulser machine allowing tension and compression fatigue loading and the extensometer SHOWA-SOKI TCK-1-IF, which was used for strain measurements. The gauge length was 25 mm (Figure 14). All tests were performed under constant velocity of 0.1 Hz and were rate independent.

A total of 4 specimens made of S275 grade structural steel were subjected to symmetrical cyclic stress and symmetrical load of the same range ε = ±1.5, but with different previous histories.

Two test specimens were exposed to the same load regime. A total of two different load histories were applied to the specimens during the tests. Each load history consisted of blocks of 5 symmetric, full cycles (Figure 15). The load application rate was constant in all tests and was 0.1 Hz. The test regime with the number of cycles for all samples subjected to cyclic load is shown in the Table 1.

A graphical representation of two different load histories in the form of a change in dilatation ε as an input as a function of time t is shown in Figure 15.

The given load history B represents a “shortening” of the stress history A, i.e., this load history is formed based on the previous one (A → B) by omitting the first ranges. However, the last, observed ranges in the cyclic load histories are the same. A comparison of hysteresis loops of the same amplitudes, formed at different load histories, is shown in Figure 16.

Experiments have shown remarkable matching of hysteresis loops with same dilatations ± ε(t), independent of the stress history (A or B). A comparison of the numerical model with the results of the experiments is shown in Figure 17. The comparison of results was performed outside the damage zone.

Comparison of the values obtained using the Preisach model defined by Equation (11) and the results of the experiments showed negligible deviations in the zone of nonlinear response. As the deviations are small, it can be concluded that the proposed model provides satisfactory solutions for defining the behavior of structural mild steel exposed to cyclic loading.

During the experiment the surface temperature at the middle of specimen was recorded via a Flir E50bx thermal imaging infrared camera. Temperature increase due to plastic deformation is evident and it is shown in Figure 18. Results obtained in Section 3 and specifically with Equation (35) are within decimal agreement with the recorded one shown in Figure 18.

## 6. Conclusions

In this paper, the Preisach model of elasto-plastic cyclic loading of a mild construction steel S275 specimen was considered. It is shown that the Lüder dilatation disappears after first cycle. Comparison of theoretical and experimental results are evident. In addition using Preisach model the locked-in energy and heat energy are calculated. After that the temperature field within the sample is obtained. The measurements of temperature during cyclic loading were performed using a thermal imaging infrared camera Flir 50bx. The calculated and measured temperatures are almost the same.

The paper clearly shows the capability of the Presisach model to analyze all aspects and phenomena associated with cyclic loading of structures in the elastoplastic region. The results obtained in this study clearly show that the model is simple enough to solve complicated structural analysis problems of cyclically loaded parts of structures. The results and findings of this paper are also background for scientists dealing with embedded sensors such are FBG, necessary for structural health monitoring of so-called smart structures.

In addition, the experimentalists now have a theoretical basis for measurements of elastic and plastic strains under severe loadings using regenerated fiber Bragg gratings sensors, as well as using a nested split-ring resonator probe.

## Figures and Tables

**Figure 1 sensors-21-03546-f001:**
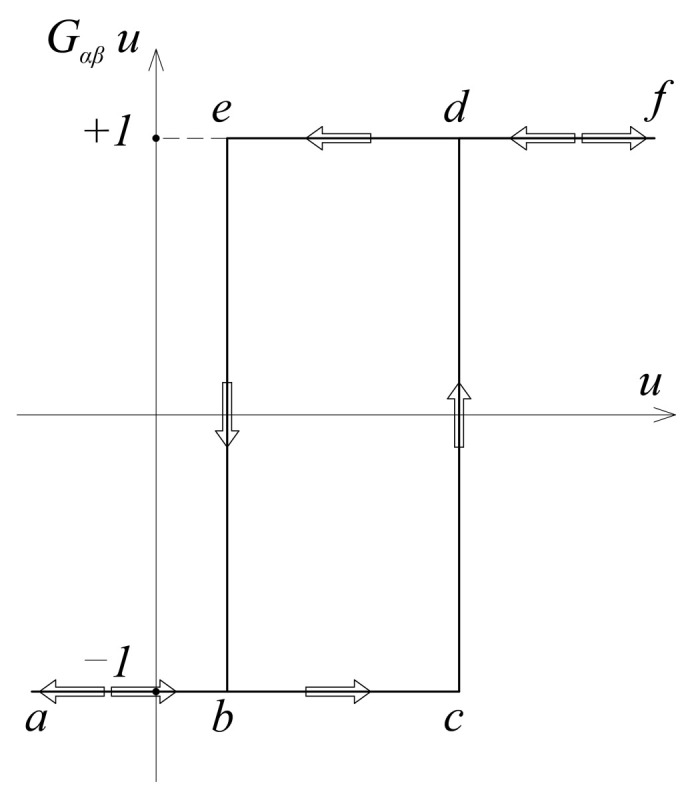
Elementary hysteresis operator.

**Figure 2 sensors-21-03546-f002:**
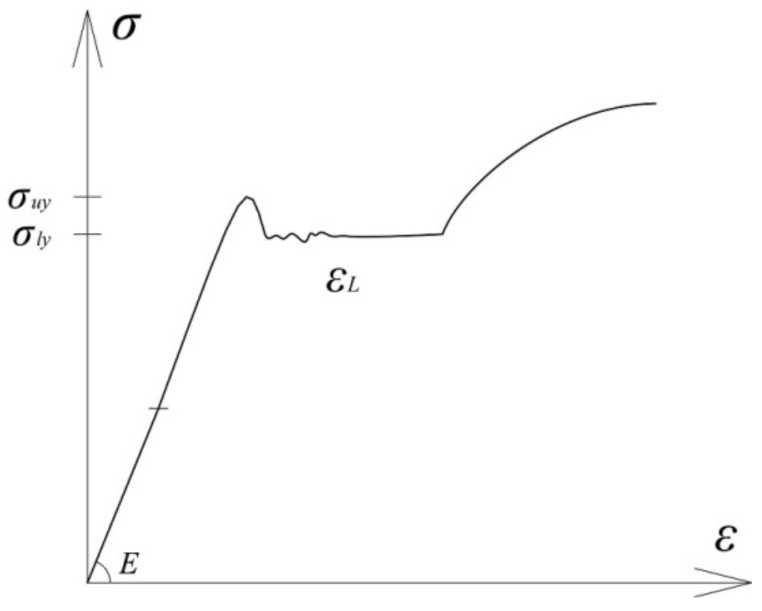
The Lüders band phenomenon occurring under monotonous loading of mild steel.

**Figure 3 sensors-21-03546-f003:**
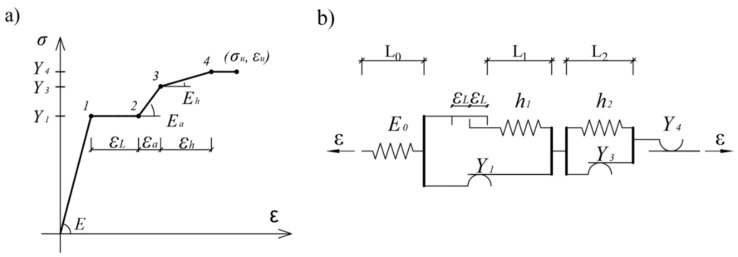
(**a**) Quintuple linear working diagram and (**b**) mechanical model.

**Figure 4 sensors-21-03546-f004:**
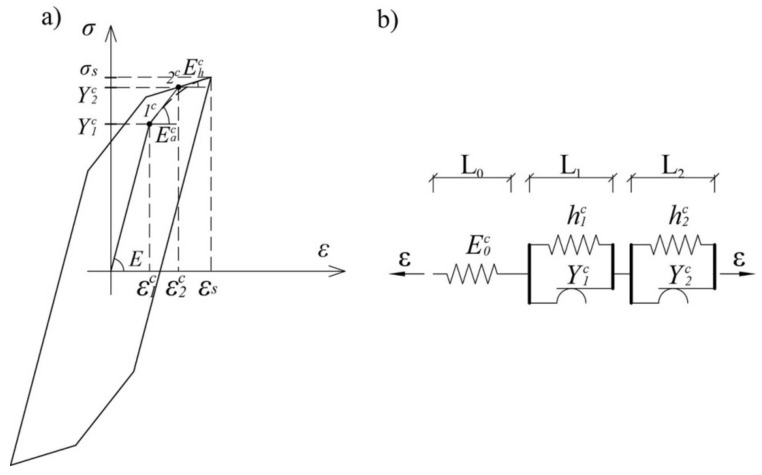
(**a**) The stress–strain diagram of structural mild single crystal steel under cyclic loading and (**b**) mechanical single crystal model.

**Figure 5 sensors-21-03546-f005:**
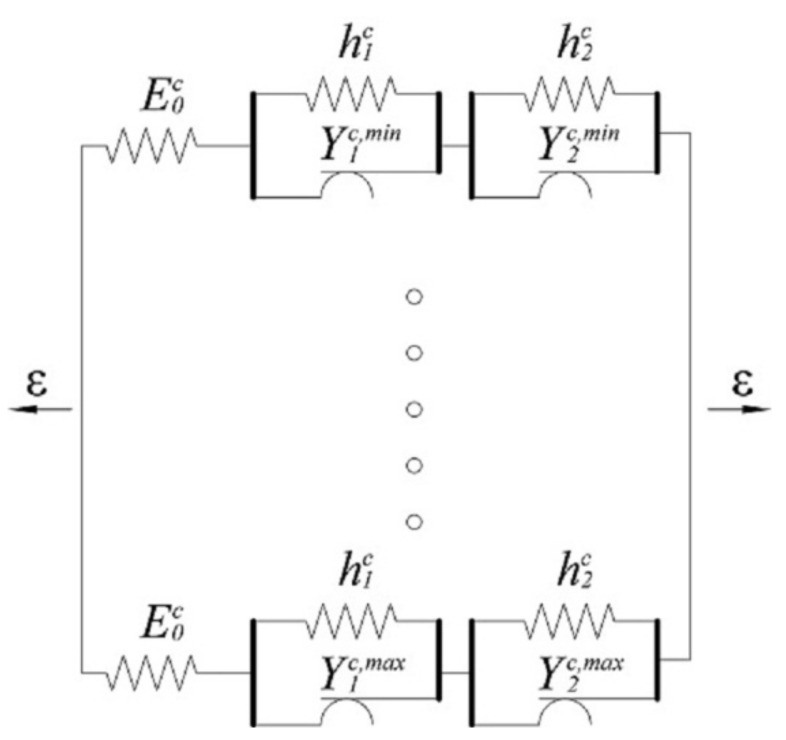
Model of polycrystalline material: parallel connection of an infinite number of single elements with yield limits *Y_i_^c,min^* ≤ *Y_i_* ≤ *Y_i_^c,max^*.

**Figure 6 sensors-21-03546-f006:**
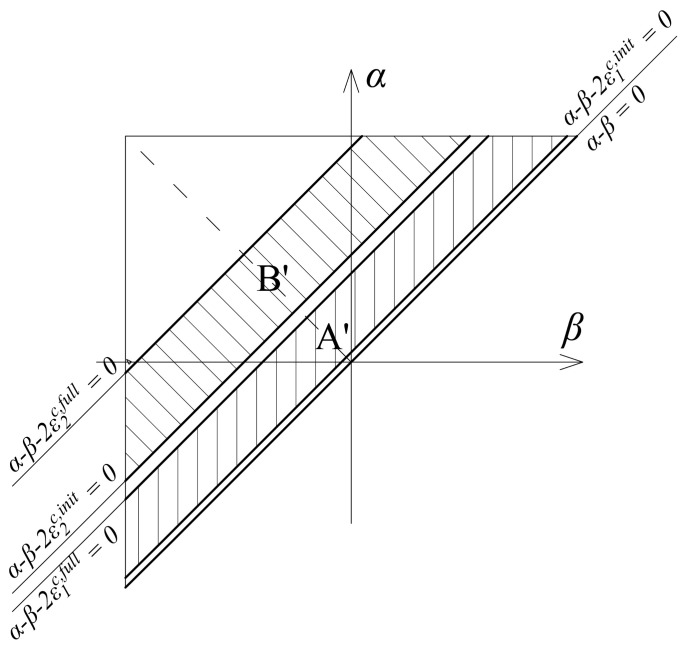
The Preisach triangle for the material model defined by Equation (11).

**Figure 7 sensors-21-03546-f007:**
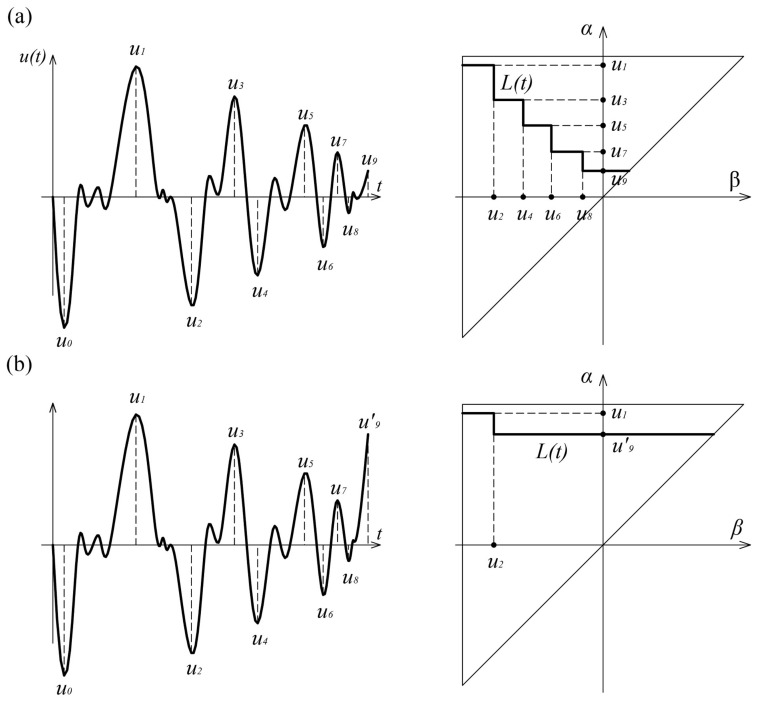
The wiping-out property of the Preisach model: (**a**) Load history *u_1_–u_9_*; (**b**) Load history *u_1_–u′*_9_.

**Figure 8 sensors-21-03546-f008:**
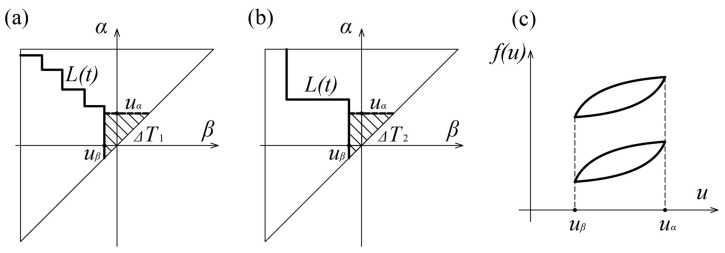
The congruency property of the Preisach model: (**a**) *α*–*β* diagrams for thr input *u*_1_; (**b**) *α*–*β* diagrams for thr input *u*_2_; (**c**) The coincidence for loops for input *u*_1_ and *u*_2_.

**Figure 9 sensors-21-03546-f009:**
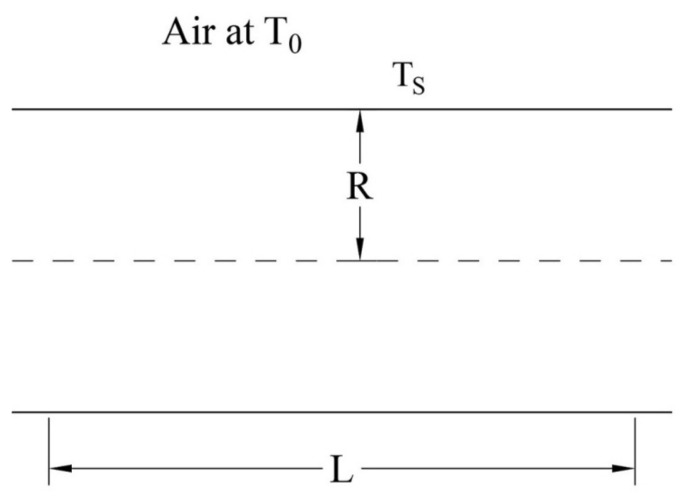
Geometry sketch of a cylinder.

**Figure 10 sensors-21-03546-f010:**
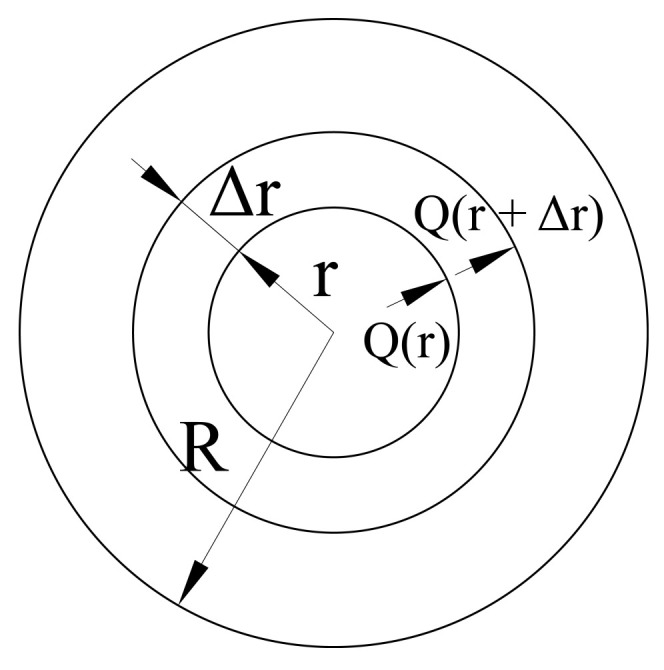
Distribution of flux.

**Figure 11 sensors-21-03546-f011:**
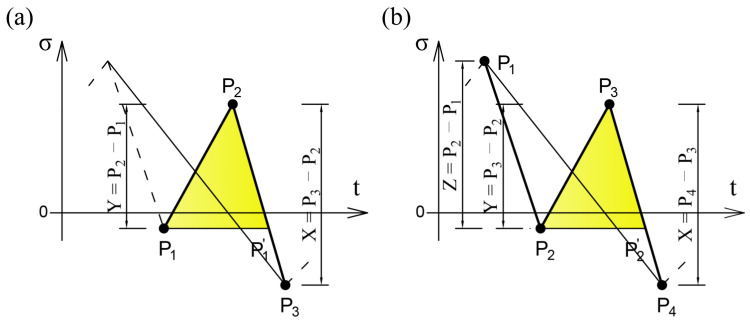
Rainflow cycle counting method: (**a**) three-points algorithm and (**b**) four-points algorithm.

**Figure 12 sensors-21-03546-f012:**
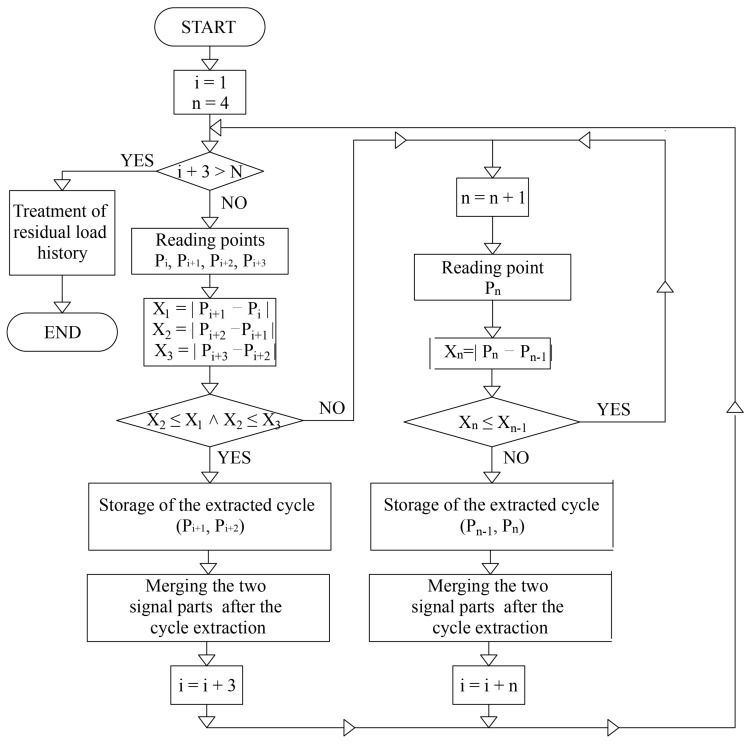
Rainflow method algorithm with the n-points criterion.

**Figure 13 sensors-21-03546-f013:**
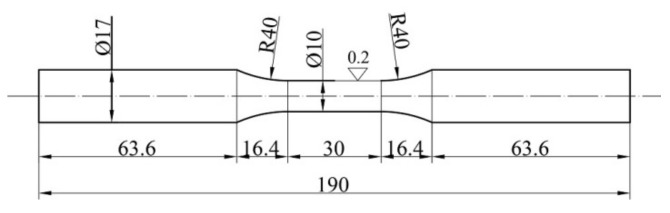
Test specimens dimensions.

**Figure 14 sensors-21-03546-f014:**
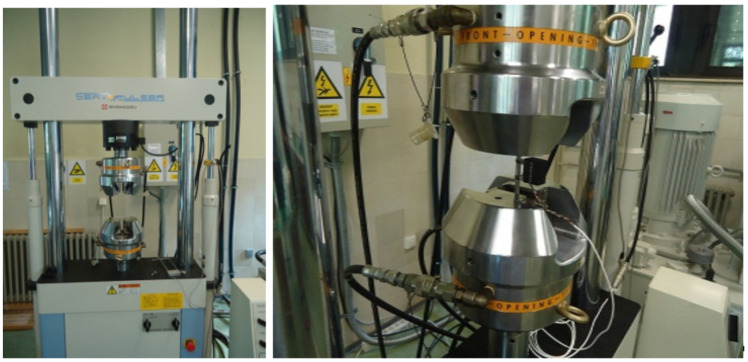
Test equipment configuration.

**Figure 15 sensors-21-03546-f015:**
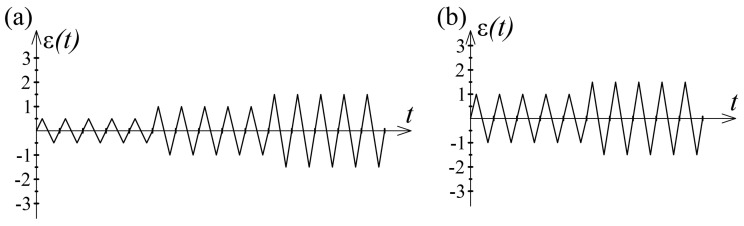
(**a**) Load history A; (**b**) Load history B.

**Figure 16 sensors-21-03546-f016:**
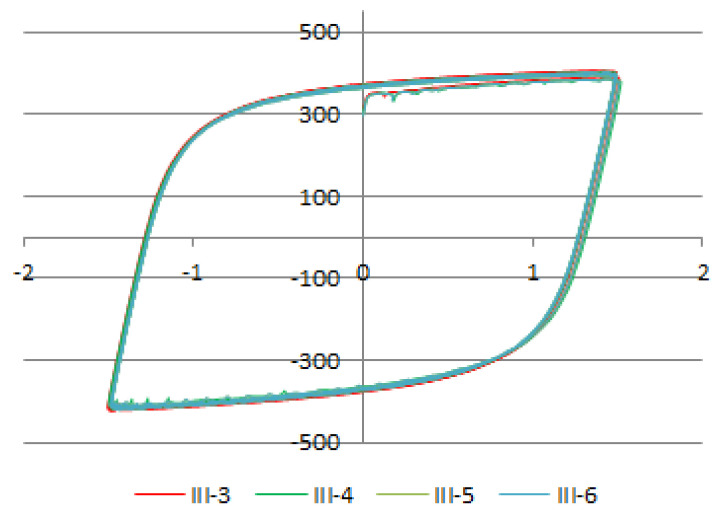
A comparison of hysteresis loops of the same amplitudes ε = ±1.5, formed at different load histories.

**Figure 17 sensors-21-03546-f017:**
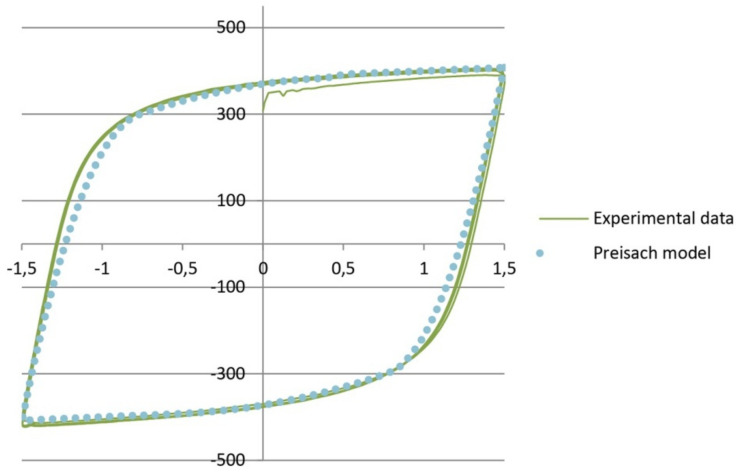
Comparison of numerical and experimental results.

**Figure 18 sensors-21-03546-f018:**
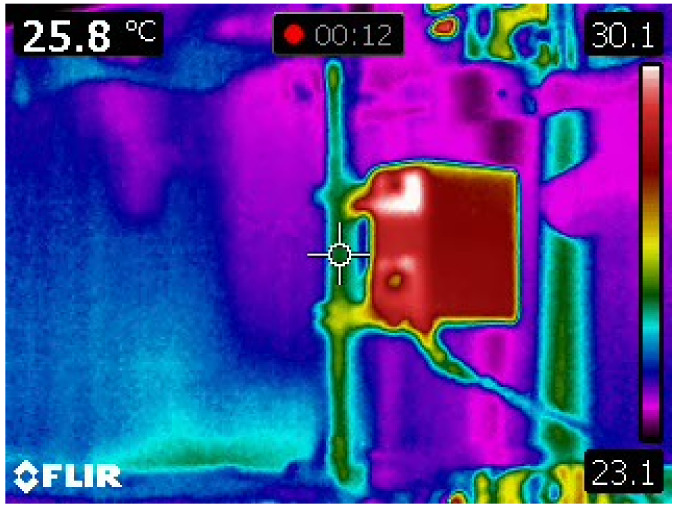
Thermographic recording of the temperature of the specimen.

**Table 1 sensors-21-03546-t001:** Sample testing regimes.

*ε* [%]		±0.5	±1	±1.5	
Number of cycles	III-3	5	5	5	A.
	III-4	5	5	5	
	III-5	/	5	5	B.
	III-6	/	5	5	

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
