# Peer review of "Preisach Elasto-Plastic Model for Mild Steel Hysteretic Behavior-Experimental and Theoretical Considerations"

_sensors, 2021, doi:10.3390/s21103546_

Round 1
Reviewer 1 Report
The purpose of the paper presented by authors is to establish the stress-strain relation and construction of cyclic stress-strain loops for arbitrary element in structures. The topic of the article is interesting and still up to date. The knowledge of the material behaviour is really very important for structures resistant to earth-quakes or bridges exposed to heavy moving loads, aircraft structures, transport means bearing frames and in generall in the situation that are exposed to cyclic loads during service life.
The authors presented results of the the Preisach model of elasto-plastic cyclic loading of mild construction steel S275 specimen investigation here. In the final comparison, the results of the theoretical analysis and experimental work show a high degree of similarity. This proves that the theoretical assumption supports the real measured findings.
The authors showed capability of their formulation of Presisach model to analyze chosen aspects and phenomena associated with cyclic loading of structures in elastoplastic region. As a positive addition, authors offer for practical use the theoretical basis for measurement of plastic strains using used descripted sensors under investigation circumstances.
The topic of the article is processed on very good level. The article express the atributes of the scientific paper.
Maybe information about the motivation of the author for this research may improve the attractivity for the readers. Why they chose the steel S275? Will the model pay reliable for other types of steels? What are the ctiteria for the parameters of steels, where these results will be valid. If it is possible to specify. What is the main success of this research? Where is possible the results utilize mainly? What is the What is fact the greatest practical success of the research?
Reviewer 2 Report
The purpose of this paper is to establish theoretical backgroud for embedded sensors like regenerated fiber Bragg gratings (RFBG) for measurement of strains and teperature in real structures. in did, the Preisach model of elasto-plastic cyclic loading of mild construction steel S275 specimen was calculated by fomula on wiping out and congruency properties, and all these analysis and formula derivation are necessary for establish theoretical backgroud for RFBG.
However, too much description of basic concepts was presented, such as: 2.1 single crystal Preisach model, Figure 1 and Figure 14 could be deleted.
The more important content was touched on lightly. such as: how to measure strain by RFBG , there was no specific explanation in this regard of detail theoretical approach.
